# Impact of mothers' and fathers' math self-concept of ability, child-specific beliefs and behaviors on girls' and boys' math self-concept of ability

**Paulina Feige**[1]*, **Rainer Watermann**[1], **Sandra Simpkins**[2], **Jacquelynne Sue Eccles**[2], **Elisa Oppermann**[3]

1 Division of Empirical Research in Education, Department of Education and Psychology, Freie Universität Berlin, Berlin, Germany, 2 School of Education, University of California, Irvine, CA, United States of America, 3 Division of Early Childhood Pedagogy, Faculty of Education, Leipzig University, Leipzig, Germany

* paulina.feige@fu-berlin.de

**Data Availability Statement:** All data files are available from the Harvard Dataverse (https://doi.org/10.7910/DVN/T1QCQQ).

## Abstract

The present study investigated the longitudinal direct and indirect relations between mothers' and fathers' math ability self-concept, their child-specific math performance expectations and encouragement of math and science-related activities at home, and girls' and boys' math ability self-concept. Structural equation models were performed with longitudinal data from three waves of the Childhood and Beyond Study (CAB). The final sample consisted of 517 children and their mothers and fathers. The majority of children attended 2nd (26.1%), 3rd (25.5%) or 5th (40.4%) grade at first measurement point. Our results suggest that mothers and fathers with higher math ability self-concepts had higher expectations of their sons and encouraged their sons more, but not their daughters. Fathers' math ability self-concept was indirectly related to the self-concept of their sons and this association was mediated by performance expectations. Furthermore, both boys and girls profited from their fathers' expectations and girls benefitted from their fathers' encouragement of math and science-related activities at home. In contrast, we found no effects from mothers' beliefs and behaviors on child's math ability self-concept. The findings underscore the relevance of fathers' educational participation in the development of the math self-concept of ability of their children.

## Introduction

Math self-concept of ability (MSC) has been demonstrated to be one of the most important constructs to explain gendered career aspirations and choice behaviors [1–3]. Specifically, studies suggest that boys and girls assess themselves differently in math as early as elementary school (e.g., [3–6]) and that these gender differences in students' MSC contribute to gender differences in individuals' educational and occupational choices [7–11]. It is widely assumed that children's MSC is shaped by information from various sources, such as teachers' beliefs

**Funding:** The Childhood and Beyond Study was funded by the National Institute for Child Health and Human Development, Grant HD17553. The funders had no role in study design, data collection and analysis, decision to publish, or preparation of the manuscript.

**Competing interests:** The authors have declared that no competing interests exist.

[12] and ability comparisons with peers or across domains [13]. Additionally, parents were thought to play a crucial role in the development of MSC in girls and boys [14]. However, research has paid little attention to the potential intergenerational transfer of MSC from mothers and fathers to their children. Furthermore, while studies on academic self-concept often differentiate children based on their gender, there is less research that differentiates mothers and fathers. In educational psychology research, studies have primarily focused on mothers or composite measure combining mothers and fathers. Therefore, little is known about whether frameworks for parental socialization work similarly for fathers and mothers. To close this gap, drawing on data of the Childhood and Beyond Study [15], this study investigates the indirect effects between mothers' and fathers' MSC and their children's MSC via parents' performance expectations and math- and science-related encouragement, and the differential relations for boys and girls.

## Mothers' and fathers' beliefs and behaviors as predictors of girls' and boys' MSC

Because parents are considered to be the most important agents of primary socialization [16], several theories emphasize the importance of parental beliefs and behaviors for the development of school-relevant outcomes in children (e.g., [17, 18]). For example, within the *Bioecological Model of Human Development* [17], the immediate social environment, the so-called microsystem, provides a crucial context for human development. Especially in the early years, parents are considered the principal persons within the microsystem; however, over time, interactions with *significant others*—such as teachers and peers—also influence a child's development. While acknowledging especially the impact of teacher's beliefs and behaviors on a child's MSC [19, 20] this paper focuses on parents as the primary socializing agents. The *Parental Socialization Model* [21] provides a broad theoretical overview of how parents can affect their children's beliefs and behaviors. In our study we depict three components of the model: Within the model, parental MSC is a *general parental belief* that directly and indirectly affects children's learning- and achievement-related motivational beliefs via *parental child-specific beliefs* (e.g., parent perceptions of child's abilities/talents) and *parental child-specific behaviors* (e.g., parent encouragement of children to participate in activities). More specifically, it can be inferred that mothers and fathers with higher ability self-concepts may have higher performance expectations for their children's academic performance, and that these expectations translate into more frequent parental encouragement and provision of educational activities related to that domain at home, which would–in turn–affect children's motivational beliefs, like their own MSC.

## General beliefs: Mothers' and fathers' MSC

Within the *Parental Socialization Model*, parents' general beliefs are assumed to affect children's motivation directly through different mechanisms: First, in line with the social learning theory [16], parents act as role models. Since children are most likely to act like people who are similar to them [22], e.g., same-sex parents, it is thought that gender differences in children's motivation are a consequence of differential behaviors (i.e. role modeling) of mothers and fathers [21]. Second, parents may communicate their self-beliefs more directly to their children. By incorporating the messages mothers and fathers provide about their own self-perceptions, boys and girls may develop differential self-beliefs [21]. Taking into account that both girls and mothers have lower MSC than boys and fathers, respectively (e.g., [3, 6, 23]), one could assume that there is a gendered intergenerational transmission of paternal and maternal beliefs regarding mathematics. By parents passing on their own self-beliefs directly or through

exemplary behavior, children might learn what behaviors and beliefs are considered appropriate according to their gender roles, making it more likely that girls will adopt their mothers' beliefs and boys their fathers' beliefs [23].

However, despite these theoretically assumed mechanisms, the relation between mothers'/fathers' and girls'/boys' MSC has received little attention in research. To our knowledge, only two studies explored the direct link, with somewhat mixed findings. Eccles- Parsons et al. [23] found only weak evidence for a relation between mothers' and fathers' MSC and girls' and boys' MSC from fifth to eleventh grade. Theurer et al. [24] found that parental MSC at the end of second grade longitudinally predicted children's MSC one year later. In a separate gender analysis, only girls' MSC were predicted by their parents' MSC. Because the questionnaires were largely completed by the mothers (75% mothers, 20% both parents, 7% fathers), the authors [24] concluded that these relations might be stronger in same-sex dyads.

## Child-specific beliefs: Mothers' and fathers' performance expectations

In addition to the direct links between parents' general beliefs and children's motivation, the Parental Socialization Model [21] suggests an additional, indirect link through parents' beliefs about their child, such as performance expectations. Specifically, the model posits that parents' general beliefs—such as their own MSC—influence what they expect from their children. These expectations, in turn, shape their children's MSC. Regarding the link between parents' expectations and the MSC of their children, two mechanisms are expected: First, parents may provide varied learning opportunities based on their beliefs about their child's abilities, such as domain-specific encouragement (as discussed in the following section). Second, parents often communicate these beliefs directly through messages about their child's abilities, helping the child interpret their own experiences. As a result, children seem to adopt their parents' interpretations of reality and integrate them into their own self-related cognitions [21]. In recent decades, extensive research has shown that there is a positive relation between mothers' and fathers' child-specific math performance expectations and children's MSC for different samples with respect to age and nationality [12, 17, 23, 25–28]. For instance, Pesu et al. [26] found in a Finnish sample that mothers' and fathers' beliefs about their child's abilities in 7th grade predicted adolescents' mathematics self-concept (MSC) in 9th grade, even after controlling for previous MSC and mathematics performance. Similarly, Tiedemann [28] found in a German sample of 3rd and 4th graders that, independent of prior math grades and teachers' perceptions of ability, parents' beliefs about their child's abilities in math predicted the child's MSC. In a US sample, Gniewosz et al. [29] found, that the relation between mothers' beliefs about their child's mathematics competence and the child's MSC increased during the transition to secondary school, while the effect of grades decreased. Unlike grades, parental performance expectations remained more stable across time, leading the authors to conclude that students perceive parental competence beliefs as more reliable, thus placing greater emphasis on them [29].

Furthermore, a sizeable proportion of studies provide evidence that parental math related beliefs are gender biased. For example, studies suggest that parents of daughters rate the difficulty of mathematical tasks higher and expect lower performance even when controlling for actual performance than parents of sons [23, 30]. It is widely assumed that these gender biased parental beliefs are powerful predictors of the differences in boys' and girls' MSC [14, 31]. With respect to parental gender, studies suggest that both mothers' and fathers' performance expectations are related to children's MSC [5, 23, 26, 28].

Given the empirical evidence suggesting that parental expectations shape children's MSC, it is crucial to investigate the factors that might affect parental expectations, like parents' own

MSC. For example, it could be assumed that parents may generalize their own beliefs into their expectations of their child's performance. Research findings on the relations between parents' own self-concept of ability and their child-specific performance expectations are scarce. As the only study we are aware of, Dickens and Cornell [32] found an indirect association between mothers' and fathers' and daughters' MSC via mothers' and fathers' performance expectations in a sample of high ability adolescent girls. Because of the selective sample of high ability girls, it is not possible to draw clear conclusions for other subpopulations of children. Thus, it remains unclear whether the impact of parents' MSC on children's MSC is mediated by parents' child-specific performance expectations.

## Child-specific behaviors: Mothers' and fathers' encouragement

Several theoretical frameworks emphasize the importance of the home learning environment, such as parental encouragement of domain-specific activities, for the development of children's beliefs and educational outcomes [18, 21, 33]. Domain-specific parental encouragement encompasses a wide range of support for home learning and was found to be associated with higher achievement among children [34, 35]. According to the *Parental Socialization Model* [21], parents' domain-specific behaviors, like their encouragement, in turn, depend on parents' general and their child-specific beliefs. Specifically, parents who have higher domain-specific performance expectations for their children are expected to provide their children with more domain-specific home learning opportunities [21]. In turn, these learning opportunities are theorized to influence children's self-concept of ability because these learning opportunities signal which domains parents consider important and appropriate, as well as influence children's skill development in specific domains [36]. This is particularly relevant as research indicates parental behaviors are gender biased with parents of sons providing more learning opportunities in math than parents of daughters [30, 36].

Past studies support the relations postulated above [37–40]. For instance, Simpkins et al. [40] found for the sample of the CAB Study, that mothers' general (e.g., importance of the domain) and child-specific beliefs in math (e.g., perception of child's ability) predicted their behaviors (e.g., modeling, encouragement, coactivity) in math. In turn, these behaviors predicted the motivational beliefs of the adolescents one year later (e.g., task value, self-concept of ability). Furthermore, they found that mothers' behaviors mediated the relations between mothers' and children's beliefs.

Although this pathway is theorized, the link between mothers' and fathers' own MSC and their math-specific encouragement has not been investigated yet. For the domain of sports, Harold et al. [41] found that parents (mothers and fathers combined) who perceived themselves as having more aptitude in sports, provided more opportunities for their children to participate in sports. The relation between parental self-concept and their behaviors in other domains remains to be determined.

Largely unresolved is the question of the extent to which maternal and paternal encouragement of domain-specific activities differentially impact daughters and sons. For example, four of the studies cited above did not distinguish between parental gender [37–40]. Since the samples used were predominantly mothers, previous research largely illustrates how maternal behaviors influence children's motivational beliefs. The work by Simpkins et al. [27] took the differential impact of fathers and mothers into account. Here, they found that paternal behaviors had an impact on children's MSC in elementary school, but maternal behaviors did not. However, it remains unclear, how paternal and maternal behaviors affect boys' and girls' MSC beyond elementary school.

### The current investigation

To close the research gaps mentioned above, this study investigates the indirect relations between mothers' and fathers' MSC and boys' and girls' MSC, mediated by parents' child-specific math performance expectations, and parents' encouragement regarding math and science activities. Though previous studies have mostly focused on either the relations between parents' child-specific beliefs and children's MSC or on the relations between parents' behaviors and children's MSC, the present study examines the entire chain of relations between parents' MSC and children's MSC as theorized in situated expectancy-value theory [2]—thereby accounting for the potential emergence of gender differences at multiple points. Hence, we tested a mediation model that encompassed parents' MSC, parents' child-specific math performance expectations and their encouragement regarding mathematical activities and children's MSC.

Because studies suggest that parental beliefs and behaviors in early and middle childhood may have long-term effects on children's academic outcomes (e.g., [39, 40]), the present study aims to gain insights into both the short- and long-term effects. To accomplish this purpose, we look at the impacts of parental beliefs and behaviors in elementary school on children's self-concepts in both elementary (w2, one year later) and secondary school (w3, five years later).

Based on the theoretical and empirical research literature, we formulate the following hypotheses: First, in the cross-sectional part of the model, we hypothesize that the MSC of mothers and fathers are positively related to their encouragement of math-related activities via child-specific performance expectations in mathematics.

Second, in the longitudinal part of the model, we examine the relations between mother's and father's MSC and the child's MSC in w2 and w3. We hypothesize that mother's and father's MSC are indirectly related to their children's MSC via performance expectations and encouragement. In addition, we assume an indirect effect of mothers' and fathers' MSC on the child's MSC via performance expectations via encouragement (serial mediation).

Third, we are interested in the gender-specific relations and whether differential findings emerge within parent-child dyads differentiated by gender. Here we expect that parental and child's MSC is stronger related in the same-sex dyads than in the opposite-sex dyads.

Three control variables were included in the analysis: First, based on previous findings that parents' achievement-related beliefs and behaviors are positively related to parental education [37, 42], the study considers parental education as a potential confounder. Second, as parenting behaviors differ as children mature [43], we additionally controlled for cohort status. Third, as we are aware of the positive correlation between math grades and children's MSC [44], we included math grade as a control variable in the model. Paths of the control variables were applied to all constructs of the model.

## Method

### Participants and procedure

We draw on data from 517 children and their mothers and fathers from all three cohorts of the CAB Study. CAB is a large-scale longitudinal study that included 10 data collection points, which began in the school year 1986/1987 in four school districts in Southeastern Michigan with three different age cohorts. Recruitment of participants started on December 1st, 1986 and ended October 30th, 1988. Parents provided written informed consent prior to their children's participation. A letter outlining the study details was sent home with the children, which included an informed consent form for the parents to sign. The signed consent forms

were then returned via the children to their teachers. As an incentive for participation, the researchers informed the schools that the population-level results would be shared with them, and that each participating teacher/classroom would receive a small financial contribution, which could be used to purchase educational materials. Data collection was carried out in the classrooms only for those students whose parents had returned the signed consent form. The data was collected through interviews and questionnaires, although only the questionnaire data were used for the present study [15]. All questionnaires, additional information on the study design and previous publications can be found at https://garp.education.uci.edu/cab. html. The Ethics Committee of Freie Universität Berlin has confirmed that no ethical approval is required. To examine the parental influences on the development of their children's MSC, we used parent report measures at wave 1 (1988/1989; parents' measures were only collected at wave 1), and child report measures at waves 2 (1989/1990), and 3 (1993/1994). Between wave 2 and 3 the children changed to secondary school. Only children who participated in w1 in the study and who had at least one parent (mother or father) participating in w1 were included in our study ($n$ = 517). Of the total sample ($N$ = 1169), 999 children and 517 parents took part in w1. To identify possible systematic biases in the results, the children in our study were compared with the excluded sample ($n$ = 482) using $t$-tests for MSC and math grade. The results indicate that the included sample had significantly higher grades at w1 (wave 1) and reported a higher MSC at w3 (S2 Table). The effect sizes of the mean differences were small, but when interpreting the results it must be considered that our study sample is positively selected.

The study sample consisted of 49.9% females and the mean age of the students at w1 (Spring 1989) was 9.94 years ($SD$ = 1.41). At w1, the majority of the students attended 2nd (26.1%), 3rd (25.5%) or 5th (40.4%) grade. The sample was predominantly White (94.6%). Mothers and fathers reported the most common categories regarding their highest educational attainment as follows: "grade school" (0.2% mothers, 0.6% fathers), "some high school" (1.4% mothers, 0.8% fathers), "high school graduate" (16.8% mothers, 8.5% fathers), "some college or technical school" (30.8% mothers, 25.1% fathers), "associate's degree" (10.3% mothers, 7.2% fathers), "college graduate" (15.7% mothers, 20.3% fathers), "some graduate work" (11.4% mothers and 8.1% fathers), "master's degree" (7.7% mothers, 13.5% fathers), Ph.D. (0.2% mothers, 2.1% fathers) and "Advanced professional degree" (0.6% mothers, 3.5% fathers). In w1, 0.6% of mothers stated that they were single, 89.7% were married, 1.7% were separated, 5.0% were divorced, and 0.6% were widowed.

## Measurement instruments

**Parents' MSC.** Parents' MSC was measured by a single item at w3. The instruction was as follows: "We are interested in how you would describe yourself with respect to the qualities listed below. Please rate each quality in terms of how true it is of you most of the time. The larger the number, the more you think you possess that quality." To measure mathematical self-concept, the item "Good at math" was given in the list. Parents rated their response on a 7-point Likert scale (1 = *Not at all true* to 7 = *Very true of me*).

In large-scale studies, such as the CAB study, single-item measurement is particularly useful due to time constraints [45, 46]. Furthermore, results of Gogol et al. [47] suggested that single-item measures of MSC meet both reliability and validity criteria.

**Parents' child-specific mathematical performance expectations.** Parents' child-specific mathematical performance expectations were operationalized using the item "How well do you think this child will do in each of these areas in the next year? Please use this scale: -Math". Responses could be given on a 7-point Likert response format from *Not at all well* (1) to *Very well* (7). With regard to the criterion validity of the item, correlation patterns with grades are

comparable to those found in Pesu et al. [26] with multi-item scales (Pesu et al. [26]: $r_{mother}$ = 41, $r_{father}$ = 31; in our study: $r_{mother}$ = 36; $r_{father}$ = 31, S3 Table).

**Parents' child-specific encouragement of math and science-related activities.**  Parents' encouragement of math and science-related activities was measured by one item at w3. After the instruction "Parents can influence their children's interests by either encouraging or discouraging various activities or interests. Please indicate below extent to which you <u>encourage</u> the following activities for <u>this</u> child", they could answer the item "Doing math-related (e.g., math-oriented games such as mastermind) or science-related (e.g., chemistry sets) activities at home"on a 7-point Likert response format from *strongly discourage* (1) to *strongly encourage* (7).

**Child's MSC.**  Child's math self-concept was measured at w3, w4, and w5 by a scale consisting of five items ($\alpha_{w3}$ = .76; $\alpha_{w4}$ = .84; $\alpha_{w5}$ = .90): "How good at math are you?"(response format: 1 = *not very good*, 7 = *very good*); "If you were to list all the students from best to worst in math where are you?"(1 = *one of the worst*, 7 = *one of the best*); "Compared to other subjects how good are you at math?" (1 = *a lot worse*, 7 = *a lot better*); "How well do you expect to do in math this year?" (1 = *not well*, 7 = *very well*); "How good would you be at learning something new in math?" (1 = *not very good*, 7 = *very good*).

**Parental education.**  To measure parental education, mothers and fathers indicated their highest level of educational attainment on a list of precoded responses from *grade school* (1) to *Advanced professional degree* (10).

**Math grade.**  Math Grade was collected from school records at w3. The grades were coded from *F* (1) to *A+* (16).

**Cohort status.**  Two dummy variables were created to account for cohort status. As most of the children were in the oldest cohort, we used this cohort as the reference group. The first dummy variable coded 1 for a child in the youngest cohort, and the second coded 1 for a child in the middle cohort.

## Statistical analysis

Descriptive statistics and *t*-tests were conducted using the scale scores of children's MSC and the single items of parental variables. Data were analyzed using M*plus* 8 [48]. Before conducting SEM, we performed multiple-group confirmatory factor analyses to examine invariance of children's MSC measures across groups (boys and girls) and across time (w3-w5; [49]). Accordingly, we first tested whether the model is acceptable in each group. Second, we tested the equal form (configural invariance) to ensure that the same number of factors and indicator-factor assignments were assessed for boys and girls and for all measurement points. Third, we tested invariance of factor loadings across gender and across time, to establish metric invariance. Metric invariance is needed to compare the latent variances and covariances. To evaluate measurement invariance, we examined the change in CFI ($\Delta$CFI). Accordingly, the change should be smaller or equal to -0.01 [50].

To test our hypotheses, we conducted a serial mediation model. To examine the effects of the mediators, we used 95% bootstrapping confidence intervals with 1000 iterations. BCBOOTSTRAP option was used to get bias-corrected confidence intervals [51]. To ensure that the lower CI is clearly away from zero, two decimal places were taken into account in the evaluation of significance (e.g. a lower CI with 0.001 and a positive upper CI was considered insignificant). The model included parental education, cohort status, and math grade as control variables. Correlations between residuals of identical items over time were allowed. The model assumed equal factor loadings across all four groups and across time for the manifest indicators of children's math self-concept of ability. The structural equation models (SEM) were conducted, computing multigroup SEM with ML Estimator.

The percentage of missing values are 0%, 6.57% and 22.63% for children's MSC at w1, w3 and w3 respectively. The percentage of missing values for mothers varied between 3.2 and 4.26%; for fathers between 32.69 and 35.02%. Model parameters were estimated using the FIML algorithm [52]. Studies suggest that FIML outperforms many techniques for handling missing data (e.g. listwise deletion) when the percentage of missing values is relatively high (e.g. 50%; [52, 53]).

To better substantiate the unique contributions of fathers and mothers on both boys and girls, our hypotheses were tested in a multi-group model with four groups: fathers/sons, mothers/sons, fathers/daughters, and mothers/daughters. Because we expect that parental and child's MSC is stronger related in the same-sex dyads than in the opposite-sex dyads (father-son vs. mother-son and father-daughter vs. mother-daughter), we constructed a variable measuring the difference in indirect effects between groups and tested this for significance using bootstrap 95% CIs (for a detailed description see [54]). A significant difference variable (hereinafter referred to as $B_{diff}$) thus indicates a significant difference between the groups.

According to Kline [55] the model fit was assessed using the following indices: $\chi^2$/df, root mean square error of approximation (RMSEA) and its 90% confidence interval, comparative fit index (CFI), and standardized root mean square residual (SRMR). Generally, cutoff values close to .06 for RMSEA, .95 for CFI, and .08 for SRMR are considered indicators of good model fit [56].

## Results

### Descriptive results

Table 1 provides gender-specific means, standard deviations, and bivariate correlations. Two-tailed $t$-tests were conducted to test descriptive differences for significance. The results revealed that boys reported higher MSC than girls. These differences were significant for w1 ($M_{Age}$ = 9.94 years, $SD$ = 1.41), $t(515)$ = -3.44, $p < .001$; and w2 ($M_{Age}$ = 10.94 years, $SD$ = 1.41), $t(481)$ = -3.63, $p < .001$. The standardized effect size for these differences was $d$ = -.30, 95% CI for $d$ [-.48, -.13] for w1, and $d$ = -.33, 95% CI for $d$ [-.51, -.15] for w2. In contrast, the difference at w3 ($M_{Age}$ = 14.94 years, $SD$ = 1.41) was not statistically significant, $t(398)$ = -1.24, $p = .215$, $d$ = -.12, 95% CI for $d$ [-.32, .07], indicate that boys' and girls' MSC converge with increasing age. With regard to children's math grade, we found no significant differences

**Table 1.  Means, standard deviations, and bivariate correlations for girls (above the diagonal) and boys (below the diagonal).**

| Variable | Girls M | Girls SD | Boys M | Boys SD | 1 | 2 | 3 | 4 | 5 | 6 | 7 | 8 | 9 |
|---|---|---|---|---|---|---|---|---|---|---|---|---|---|
| 1 MSC Mother | 4.67 | 1.72 | 4.47 | 1.74 | | -.15* | .13 | .08 | .06 | .10 | .10 | .14* | .15* |
| 2 MSC Father | 5.32 | 1.42 | 5.32 | 1.30 | 18* | | -.04 | .13 | -.04 | .03 | .00 | .04 | .00 |
| 3 Expectations Mother | 5.98 | 1.07 | 5.98 | 1.12 | .22** | .09 | | .57*** | .01 | .12 | .31*** | .26*** | .33*** |
| 4 Expectations Father | 5.77 | 1.13 | 5.95 | 1.01 | .14 | .31*** | .54*** | | .09 | .16* | .34*** | .38*** | .38*** |
| 5 Encouragement Mother | 4.04 | 1.61 | 4.38 | 1.57 | .12 | .18* | .24*** | .11 | | .28*** | .01 | -.05 | .02 |
| 6 Encouragement Father | 4.11 | 1.48 | 4.33 | 1.51 | .08 | .30*** | .21** | .32*** | .29*** | | .04 | .14 | .28*** |
| 7 MSC Child w1 | 5.34 | 0.99 | 5.65 | 1.01 | .11 | .14 | .43*** | .29*** | .05 | .0 | | .39*** | .36*** |
| 8 MSC Child w2 | 5.26 | 1.07 | 5.61 | 1.07 | .13* | .15 | .44*** | .44*** | .08 | .17* | .50*** | | .46*** |
| 9 MSC Child w3 | 4.92 | 1.05 | 5.06 | 1.20 | .08 | .15 | .33*** | .34*** | -.01 | .09 | .32*** | .32*** | |

The scale scores of child's math ability self-concept and the single items of parental variables were used. MSC = math self-concept of ability.

*$p < .05$

**$p < .01$

***$p < .001$.

between girls ($M$ = 11.37, $SD$ = 2.16) and boys at w1 ($M$ = 11.37, $SD$ = 2.2), $t(427)$ = .007, $p$ = .994, $d$ = .00, 95% CI for $d$ [-.19, .19].

We found no significant differences between fathers', $t(494)$ = -1.61, $p$ = .109, $d$ = -.18, 95% CI for $d$ [-.39, .04] and mothers', $t(334)$ = < .000, $p$ = .999, $d$ = .00, 95% CI for $d$ [-.18, .18], performance expectations of their sons compared to their daughters. Both parents reported encouraging their sons more to engage in mathematical and science activities than their daughters. The difference was significant for mothers, $t(493)$ = -2.4, $p$ = .017, $d$ = -.22, 95% CI for $d$ [-.39, -.04]; but not for fathers, $t(333)$ = -1.38, $p$ = .168, $d$ = -.15, 95% CI for $d$ [-.37, .06]. In addition, we applied paired sample $t$-tests to compare mothers' and fathers' MSC, their performance expectations and encouragement. Fathers reported higher MSC at w1 ($M$ = 5.30, $SD$ = 1.36) than mothers ($M$ = 4.63, $SD$ = 1.78). This difference was statistically significant, $t$ (331) = -5.10, $p$ < .001, with a small to medium effect size, $d$ = -.28, 95% CI for $d$ [-.39, -.17]. Additionally, compared to fathers, mothers reported having higher expectations for their daughters' math performance, $t(155)$ = 2.51, $p$ = .013, with a small effect size, $d$ = .20, 95% CI for $d$ [.04, .36]. In contrast, no difference was found between mothers and fathers in how much they encouraged their daughters, $t(156)$ = 1.24, $p$ = .218. For sons, there were no significant differences between mothers and fathers regarding either their performance expectations, $t(162)$ = 0.69, $p$ = .493, or their encouragement, $t(158)$ = 0.81, $p$ = .421.

## Parent-child models

Prior to conducting the parent-child models, we performed multiple-group confirmatory factor analyses with child's MSC measures to examine invariance across groups and across time (S1 Table). The results of models estimated on each group separately suggest the models fit the data well. Model 2 assumed invariant factor loadings across groups, and Model 3 also included invariant factor loadings across time. Both models were compared with Model 1 (equal form). As we found no decline in the CFI value between the models, we can assume metric invariance across groups and across time.

Our final SEM showed acceptable model fit [$\chi^2(704)$ = 1064.88, $p$ < .001; CFI = .95; RMSEA = .05, 90% CI (0.04, 0.05); SRMR = .06]. Note that parental effects on the child's MSC at w2 and w3 were each estimated while controlling for the child's previous MSC. Thus, all parental effects can be interpreted as effects on the child's MSC, independently of the child's prior MSC. Model results are illustrated in Tables 2 and 3.

Model results for math grade and parental education can be found in the S3 Table.

**Father-son and mother-son model.**   The unstandardized model results for sons are presented in Fig 1.

Unstandardized and standardized coefficents were also reported in Table 2, indirect effects can be found in Table 3. In the cross-sectional part of the model (all measures at w1)—holding sons' MSC and the covariates constant—fathers' and mothers' MSC was positively associated with their performance expectations. This indicates, that parents who had a higher MSC, also had higher expectations of their sons' math performance. Fathers', but not mothers' MSC predicted their encouragement. Performance expectations of fathers and mothers in turn showed a positive effect on their encouragement. As expected, the indirect effect of mothers' and fathers' MSC on encouragement via performance expectations proved to be significant (Table 3). Furthermore, we found no significant cross-sectional relation between mothers' and fathers' MSC and sons' MSC at w1. Fathers' and mothers' performance expectations, but not their encouragement was related to the MSC of their sons' at w1.

In the longitudinal part of the model, we found no direct relations between fathers' and mothers' MSC and their sons' MSC at w2 and w3. Performance expectations of fathers, but

**Table 2. Unstandardized and standardized path coefficients on children's mathematical self-concepts.**

| | Father-son | | | Mother-son | | | Father-daughter | | | Mother-daughter | | |
|---|---|---|---|---|---|---|---|---|---|---|---|---|
| | UNSTD | SE | STD | UNSTD | SE | STD | UNSTD | SE | STD | UNSTD | SE | STD |
| Stability Coefficients | | | | | | | | | | | | |
| MSC Child w1 => MSC Child w2 | 0.47*** | 0.10 | .44*** | 0.46*** | 0.09 | .43*** | 0.29** | 0.10 | .26** | 0.34** | 0.10 | .31** |
| MSC Child w1 => MSC Child w3 | 0.30* | 0.13 | .25* | 0.25* | 0.12 | .21* | 0.14 | 0.10 | .12 | 0.16 | 0.11 | .13 |
| MSC Child w2 => MSC Child w3 | 0.08 | 0.15 | .07 | 0.11 | 0.14 | .10 | 0.25** | 0.09 | .24** | 0.34*** | 0.09 | .32*** |
| Parental MSC | | | | | | | | | | | | |
| MSC F/M => MSC Child w1 | 0.20 | 0.11 | .14 | 0.19 | 0.13 | .10 | 0.07 | 0.11 | .01 | 0.22 | 0.12 | .12 |
| MSC F/M => MSC Child w2 | -0.04 | 0.07 | -.04 | 0.01 | 0.05 | .01 | 0.02 | 0.07 | .03 | 0.02 | 0.04 | .03 |
| MSC F/M => MSC Child w3 | 0.00 | 0.08 | .00 | 0.01 | 0.05 | .01 | -0.02 | 0.07 | -.03 | 0.04 | 0.06 | .05 |
| MSC F/M => Expectations F/M | 0.25*** | 0.06 | .32*** | 0.13** | 0.04 | .20** | 0.10 | 0.06 | .12 | 0.08 | 0.05 | .12 |
| MSC F/M => Encouragement F/M | 0.28** | 0.09 | .24** | 0.07 | 0.06 | .08 | -0.02 | 0.09 | -.01 | 0.04 | 0.06 | .04 |
| Performance Expectations | | | | | | | | | | | | |
| Expectations F/M => MSC Child w1 | 0.34*** | 0.09 | .33*** | 0.52*** | 0.10 | .44*** | 0.52*** | 0.10 | .44*** | 0.37** | 0.11 | .33*** |
| Expectations F/M => MSC Child w2 | 0.28* | 0.12 | .25* | 0.17 | 0.09 | .17 | 0.30** | 0.10 | .30** | 0.17 | 0.09 | .16 |
| Expectations F/M => MSC Child w3 | 0.16 | 0.15 | .12 | 0.19 | 0.12 | .16 | 0.22 | 0.11 | .21 | 0.16 | 0.01 | .14 |
| Expectations F/M => Encouragement F/M | 0.36** | 0.11 | .24** | 0.32*** | 0.09 | .23** | 0.25* | 0.10 | .20* | -0.01 | 0.09 | .00 |
| Encouragement | | | | | | | | | | | | |
| Encouragement F/M => MSC Child w1 | -0.19 | 0.12 | -.13 | -0.09 | 0.10 | -.05 | -0.04 | 0.12 | -.02 | -0.02 | 0.12 | -.01 |
| Encouragement F/M => MSC Child w2 | 0.07 | 0.06 | .09 | -0.02 | 0.04 | -.02 | 0.07 | 0.06 | .09 | -0.04 | 0.04 | -.06 |
| Encouragement F/M => MSC Child w3 | 0.09 | 0.07 | .11 | -0.05 | 0.06 | -.06 | 0.18* | 0.07 | .22** | 0.03 | 0.05 | .03 |

UNSTD = unstandardized model results, SE = standard error, STD = standardized model results, MSC = math self-concept of ability, F/M = father/mother.

[a] Arrow indicates direction of causation.

$^{*}p < .05$

$^{**}p < .01$

$^{***}p < .001$.

not mothers, showed a positive effect on the sons' MSC in w2, whereas we found no effects on the sons' MSC in w3. For encouragement, we found no effects on sons' MSC in w2 or w3 (Table 3). We found a significant indirect effect of fathers', but not mothers' MSC on sons' MSC via performance expectations at w2. No evidence for other indirect effects was found.

Our hypothesis was that parent and child MSC would be stronger related in the same-sex dyads than in the opposite-sex dyads. Thus, with respect to sons, fathers' MSC should be more strongly related to sons' MSC than mothers' MSC. To compare the effect of maternal and paternal MSC inferentially, we tested the difference between indirect effects for significance as described in the *Methods* section. At this point, however, we only report on those indirect path coefficients for differences between mothers and fathers that showed significant relations in at least one of the dyads. In the cross-sectional part of the model, the indirect path coefficent from parent MSC to encouragement via performance expecations was not significantly different in the father-son and mother-son dyads, $B_{diff}$ = -0.05, $SE(B_{diff})$ = 0.04, 95% CI for $B_{diff}$ (-0.14, 0.02). In the longitudinal part of the model, the indirect path coefficent from parental to sons' MSC via performance expecations was not significantly different for mothers and fathers, in w2 $B_{diff}$ = -0.05, $SE(B_{diff})$ = 0.04, 95% CI for $B_{diff}$ (-0.13, 0.02). In short, no statistical significant differences between mothers and fathers were found.

**Father-daughter and mother-daughter model.** The unstandardized model results of father-daughter and mother-daughter models can be found in Fig 2.

**Table 3. Unstandardized indirect effects and bootstrapping 95%-confidence intervals on children's mathematical self-concepts.**

| Path[a] | Father-son | | | Mother-son | | | Father-daughter | | | Mother-daughter | | |
|---|---|---|---|---|---|---|---|---|---|---|---|---|
| | UNSTD | CI$_{95}$ | | UNSTD | CI$_{95}$ | | UNSTD | CI$_{95}$ | | UNSTD | CI$_{95}$ | |
| | | LCI | UCI | | LCI | UCI | | LCI | UCI | | LCI | UCI |
| Effects on Encouragement | | | | | | | | | | | | |
| Total MSC F/M => Encouragement | **0.37** | 0.20 | 0.54 | 0.12 | -0.01 | 0.23 | 0.01 | -0.16 | 0.19 | -0.04 | -0.08 | 0.15 |
| via Expectations | **0.09** | 0.03 | 0.18 | **0.04** | 0.02 | 0.08 | 0.03 | 0.00 | 0.08 | 0.00 | -0.02 | 0.01 |
| Effects on MSC Child w2 | | | | | | | | | | | | |
| Total MSC F/M => MSC Child | 0.06 | -0.07 | 0.21 | 0.03 | -0.06 | 0.10 | 0.05 | -0.08 | 0.18 | 0.03 | -0.06 | 0.12 |
| Total indirect MSC F/M => MSC Child | **0.10** | 0.04 | 0.18 | 0.02 | 0.00 | 0.05 | 0.03 | -0.02 | 0.09 | 0.01 | 0.00 | 0.05 |
| Specific indirect effects | | | | | | | | | | | | |
| via Expectations F/M | **0.07** | 0.01 | 0.16 | 0.02 | 0.00 | 0.06 | 0.03 | 0.00 | 0.09 | 0.01 | 0.00 | 0.05 |
| via Encouragement F/M | 0.02 | -0.01 | 0.06 | 0.00 | -0.02 | 0.00 | 0.00 | -0.02 | 0.01 | 0.00 | -0.02 | 0.00 |
| via Expectations F/M via Encouragement | 0.01 | 0.00 | 0.02 | 0.00 | -0.01 | 0.00 | 0.00 | 0.00 | 0.01 | 0.00 | 0.00 | 0.00 |
| Effects on MSC Child w3 | | | | | | | | | | | | |
| Total MSC F/M => MSC Child | 0.08 | -0.08 | 0.20 | 0.03 | -0.07 | 0.13 | 0.01 | -0.14 | 0.19 | 0.06 | -0.05 | 0.18 |
| Total indirect MSC F/M => MSC Child | 0.08 | 0.00 | 0.17 | 0.02 | -0.01 | 0.08 | 0.04 | -0.03 | 0.12 | 0.02 | -0.01 | 0.07 |
| Specific Indirect effects | | | | | | | | | | | | |
| via Expectations | 0.04 | -0.03 | 0.12 | 0.03 | 0.00 | 0.08 | 0.02 | 0.00 | 0.08 | 0.01 | 0.00 | 0.05 |
| via Encouragement | 0.03 | -0.01 | 0.08 | 0.00 | -0.03 | 0.00 | 0.00 | -0.05 | 0.03 | 0.00 | 0.00 | 0.01 |
| via Expectations via Encouragement | 0.01 | 0.00 | 0.03 | 0.01 | -0.01 | 0.00 | 0.00 | 0.00 | 0.02 | 0.00 | 0.00 | 0.00 |
| via MSC Child w2 | 0.00 | -0.05 | 0.01 | 0.00 | -0.01 | 0.02 | 0.01 | -0.03 | 0.04 | -0.01 | -0.02 | 0.05 |
| via Expectations via MSC Child w2 | 0.01 | -0.01 | 0.04 | 0.00 | 0.00 | 0.02 | 0.01 | 0.00 | 0.03 | 0.00 | 0.00 | 0.02 |
| via Encouragement via MSC Child w2 | 0.00 | 0.00 | 0.02 | 0.00 | 0.00 | 0.00 | 0.00 | 0.00 | 0.00 | 0.00 | -0.01 | 0.00 |
| via Expectations via Encouragement via MSC Child w2 | 0.00 | 0.00 | 0.01 | 0.00 | 0.00 | 0.00 | 0.00 | -0.01 | 0.00 | 0.00 | 0.00 | 0.00 |

UNSTD = unstandardized model results, MSC = math self-concept of ability, F/M = mother/father. The *total effect* is the sum of the direct and indirect effect. The *total indirect effect* is the sum of the listed specific indirect effects. Values in bold types indicate significant effects. Please note that the bootstrapping confidence intervals do not have to be symmetrical around the parameters [57]. To ensure that the lower CI is clearly away from zero, two decimal places were taken into account in the evaluation of significance.

[a]Arrow indicates direction of causation

Unstandardized and standardized coefficents were also reported in Table 2, indirect effects can be found in Table 3. In the cross-sectional part–holding the daughters' MSC and the covariates constant–in contrast to the models of the sons, neither fathers' MSC, nor mothers' was related to their performance expectations. We also found no relation between fathers' and mothers' MSC and their encouragement. However, in line with the sons' models, fathers', but not mothers' performance expectations showed a positive effect on encouragement. This suggests that fathers, but not mothers, who had higher expectations of their daughters' performance in math also encouraged their daughters more to engage in math and science activities. The indirect effect of fathers' and mothers' MSC on encouragement via performance expectations was not statistically significant. In line with the sons' models, fathers' and mothers' performance expectations, but not their encouragement was related to the MSC of their daughters at w1.

In the longitudinal part, fathers' performance expectations showed a positive effect on the in daughters' MSC at w2. Analogous to the models of the sons, we found no other effects of performance expectations on the daughters' MSC at w2 or w3. Furthermore, we found no effects of fathers' and mothers' encouragement on the MSC of their daughters at w2. However, we found a significant positive effect of fathers' encouragement on the daughters' MSC at w3.

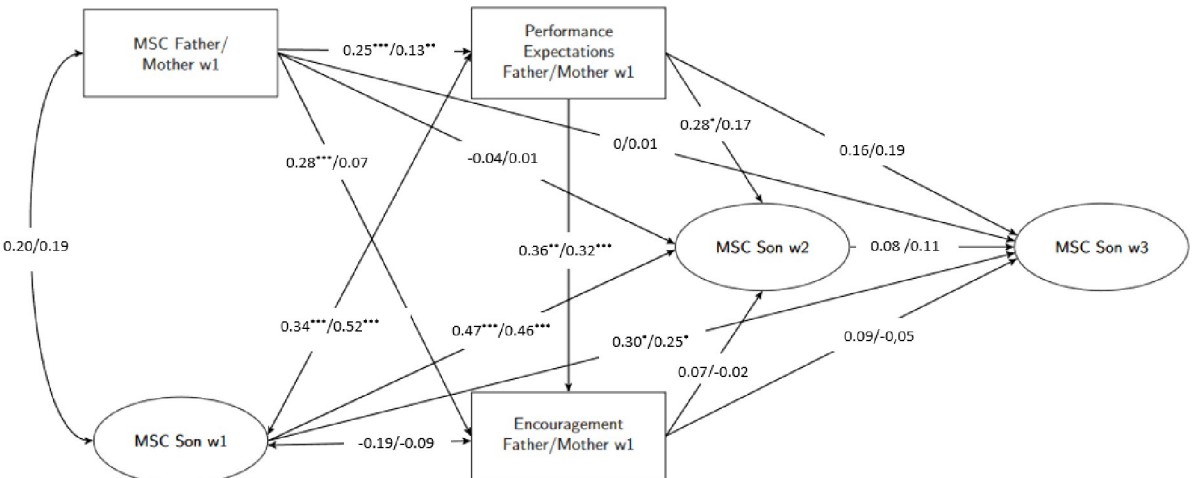

**Fig 1. Mediation model predicting sons' mathematical self-concept of ability.** Mediation model with parental math ability self-concept as predictor, performance expectations and encouragement to math- and science-related activities as mediators, and sons' math ability self-concepts at w2 and w3 as criterion, controlled for sons' self-concept at w1; unstandardized model results for fathers/mothers; paths of the control variables (cohort status, parental education, math grade) were applied to all constructs of the model; MSC: math self-concept of ability, w1: first measurement time point, w2: second measurement time point, w3: third measurement time point. $^*p < .05.$ $^{**}p < .01.$ $^{***}p < .001.$

This suggests, that when fathers encouraged their daughters to math and science activites at w1, the daughters reported a higher MSC five years later. As the lower CI of the indirect effect of mothers' MSC on daughters' MSC was 0.00, this indirect effect was considered non-significant. All other indirect effects were also not significant.

Considering that we found no significant indirect effects between mothers' and fathers' beliefs and behaviors and daughters' MSC, we did not compare mothers and fathers inferentially.

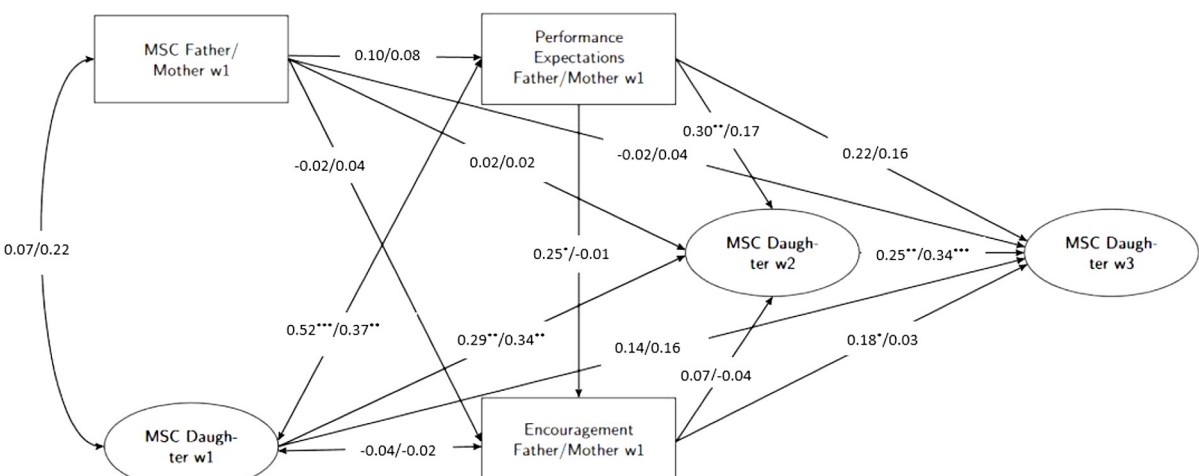

**Fig 2. Mediation model predicting daughters' mathematical self-concept of ability.** Mediation model with parental math ability self-concept as predictor, performance expectations and encouragement to math- and science-related activities as mediators, and daughters' math ability self-concepts at w2 and w3 as criterion, controlled for daughters' self-concept at w1; unstandardized model results for fathers/mothers; paths of the control variables (cohort status, parental education, math grade) were applied to all constructs of the model; MSC: math self-concept of ability, w1: first measurement time point, w2: second measurement time point, w3: third measurement time point. $^*p < .05.$ $^{**}p < .01.$ $^{***}p < .001.$

## Discussion

The contribution of this article was to pursue the question of whether mothers' and fathers' math self-concepts of ability (MSC) predicts their children's MSC and whether this relation is mediated through parents' child-related performance expectations and encouragement of math and science activities. In short, we found that fathers' and mothers' MSC were related to the performance expectations and encouragement of math and science activities of their sons, but not their daughters. Fathers' MSC was related to their sons' MSC one year later via performance expectations, no relations were found in the other dyads. Contrary to our hypothesis, we found no stronger effects in the same-sex dyads. Only fathers' performance expectations were predictive for boys' and girls' MSC. Furthermore, fathers' encouragement was predictive for girls' MSC five years later.

### Mothers' and fathers' MSC and their performance expectations and encouragement

In line with the Parental Socialization Model [21], we hypothesized that the MSC of mothers and fathers are positively related to their encouragement of math-related activities via child-specific performance expectations in mathematics. As we found significant indirect effects only in the father-son and mother-son models, we may conclude that this is especially true for sons. Parents with higher MSC had higher expectations of their sons' math performance and thus—in turn—provided more encouragement of math and science activities at home. Regarding the results for daughters, we found no direct or indirect effects of parents' own MSC on their performance expectations or encouragement. These findings contribute to the of research literature as they suggest that parents' beliefs about what they themselves are capable of in mathematics may affect parents' mathematical beliefs and behaviors' regarding their sons, but not their daughters. However, we found a positive relation between fathers' performance expectations and their encouragement regarding their daughters. Though neither parent seemed to translate their own MSC into their expectations of their daughters' math performance, fathers encouraged their daughters more to engage in math and science activities when they had higher expectations of their daughters. In line with other study results [6], parental expectations regarding their daughters' performance might be more related to other factors like perceived diligence or effort than parents' own ability beliefs. This highlights the importance of differentiating by parental and child gender when examining socialization theories.

### Relations between mothers' and fathers' MSC, their performance expectations and encouragement, and children's MSC

Following the assumptions of the *Parental Socialization Model* [21], we expected that parental MSC would be indirectly related to children's MSC one year (w2) and five years later (w3) via parental performance expectations and encouragement. Furthermore, based on former research [22, 24], we expected to find stronger effects within same-sex dyads compared to opposite-sex dyads. Our results suggest that fathers' MSC was indirectly related to their sons' MSC one year later (w1). This relation was mediated by fathers' performance expectations for their sons. We found no other significant relations between parents' and child's MSC. We can therefore only confirm our initial hypothesis that parents' MSC are related to those of their children for fathers and sons. As the effect in the father-son model was found to be positive, we conclude that fathers, who believe that they are themselves good at mathematics, also expect their sons to do well in mathematics, which in turn positively predicts their sons' MSC. However, the indirect path coefficients did not appear to differ significantly between fathers

and mothers, suggesting that the differences between fathers and mothers in their impact on their sons' MSC are negligible. Thus, we must reject the hypothesis that these effects are stronger in same-sex dyads.

**Mothers' and fathers' performance expectations and children's MSC.** Previous work [21, 23, 25, 28] found that parental performance expectations were strongly related to children's MSC. Most previous studies did not differentiate between fathers and mothers [21, 23, 25]. As an exception, Tiedemann [28] differentiated by parents' gender and found that mothers' and fathers' performance expectations were both predictive for children's ability perceptions. Here, however, no differentiation was made according to child's gender. In our study, we found that only fathers' performance expectations were predictive for sons' and daughters' MSC in w2. While maternal expectations were only cross-sectionally linked to the children's MSC in our study, it seemed to be particularly relevant for sons' and daughters' self-related cognitions in long-term, how fathers evaluate their performance. Conversely, it can also be assumed that fathers' knowledge of child's previous math's performance influences their performance expectations. However, since we controlled for prior math performance, it can be concluded that early fathers' performance expectations were predictive for children's MSC one year later beyond their own math grade. This assumption is also supported by study results suggesting that early parental math beliefs and behaviors predict children's motivational beliefs independent from their actual performance [21]. In short, our study findings imply that children construct their own MSC partly based on their fathers' beliefs; whereby, conversely, it can also be assumed that these beliefs about their children are well attuned to the abilities of their children.

**Mothers' and fathers' encouragement and children's MSC.** Based on the Parental Socialization Model, we assumed that home learning opportunities may influence children's MSC because these opportunities signal which domains parents consider important and appropriate and influence children's skill development in those specific areas [36]. In our study we investigated the relations between encouragement of math and science activities, as one component of the home learning environment, and the MSC of boys and girls. Our results suggest that only girls benefitted from their fathers', but not their mothers' math encouragement. For girls, encouragement from their father was even more strongly associated with their MSC five years later than their own prior MSC. This finding seems remarkable in view of the time interval between fathers' encouragement ($M_{age}$ of the children at w1 = 9.94) and the effect on girls' MSC ($M_{age}$ at w3 = 14.94) and emphasizes the importance of early involvement of the fathers. Simpkins et al. [27], found that multiple paternal behaviors were related to children's MSC, but no links for maternal behaviors. Here, one can consider that mothers' encouragement is more normative, and fathers who support their children in school-related matters are therefore more influential [27]. Moreover, research suggests that fathers use a wider range of cognitively challenging strategies, which may result in a more positive outcome for the children [58].

Furthermore, parental variables are only included in the model at w1, which limits the analyses to the effect of parental beliefs and behaviors at that point. As parental behaviors in particular change over time depending on their children's developmental stage [43], encouraging children to engage in math and science activities at a later stage might be more relevant to their MSC. However, there is evidence that early parental academic involvement is particularly important and that it declines during adolescence, when the peer group becomes more important [43, 59]. To test this assumption, follow-up studies should examine the influence of multiple parental behaviors at multiple times.

### Limitations and future directions

First, although we draw on a longitudinal dataset, the results cannot be interpreted in a causal manner.

Second, the sample was not randomly selected. For instance, as can be seen in the S2 Table, children in our sample were found to have better math grades compared to the excluded sample. In order to minimize the influence of these potential biases, we controlled for math grade in our analyses. In addition, we controlled for the child's previous MSC so that all parental effects can be interpreted as effects of the children's MSC, independently of their prior MSC. This approach leads to more conservative estimates, i.e., associations in our model may be underestimated. As a consequence, we argue that the longitudinal associations found in the present study can be considered meaningful even if some of the effect sizes were small.

Third, considering the study's low statistical power might possibly explain why we found significant no differences between mothers and fathers. Hence, further studies are needed with bigger sample sizes.

Fourth, it should be noted that the CAB study used data from 1988–1995, which means that the results of this study may not be fully transferable to children the present day. However, the participants in our study are now in their forties, so knowledge of their motivational beliefs during their childhood is still relevant. Furthermore, current studies still reveal gender differences in mathematical ability perceptions of boys and girls [12] and mothers and fathers [60] and parental gender stereotypic beliefs and expectations in math [3, 4, 45]. For example, Starr et al. [45] compared datasets from 1984 to 2009 and found that parents had gender stereotyped beliefs in math across data sets. Furthermore, the importance of parental performance expectations for children's MSC were also found in a current study [26]. Thus, it is still highly relevant to gain a better understanding of the early mechanism that eventually contribute to these gender difference in math-related beliefs.

Fifth, parental MSC, their performance expectations and their encouragement were measured with single items. Therefore the reliability of the constructs could not be tested. As stated in the methods section, results of Gogol et al. [47] suggested that single items measures of academic self-concepts meet both validity and reliability criteria. Furthermore, there is evidence that most single-item measures are as valid and reliable as multi-item scales [61]. With regard to performance expectations, we provided additional evidence for criterion validity in the methods section. Moreover, the significant associations of these single items with children's MSC suggest that these measure meaningful aspects of parental beliefs and practices.

Sixth, other mediators need to be captured in follow-up studies. So far, little is known about the specific situations in which general and child-specific beliefs become accessible to children. For example, one might assume that parental MSC becomes salient to children primarily in domain-specific situations, such as homework assistance or domain-specific coactivity. Further empirical work is needed to identify these specific transmission mechanisms.

Seventh, when studying effects within families, genetic factors must also be considered. Since our study assumed environmental factors within families as mediating processes between parents and child's MSC, one can also assume that there might be a genetic influence. Twin studies have shown that individual differences in self-perceived abilities can be explained by both genetic factors and non-shared environments [62, 63]. Although the data in this study did not allow us to account for a genetic component, it is important to consider the heritability of academic self-concept when interpreting our findings.

Eighth, as initially mentioned, there are a variety of processes that are involved in the development of ability self-concepts [12, 13]. In order to explore the relative importance of different processes (e.g. interactions with teachers, comparisons with peers and across domains) in the

formation of the academic self-concept, follow-up studies should focus on different mechanisms simultaneously.

Ninth, it can be assumed that maternal and paternal socialization do not operate independently, as our model implies. Thus, coordinated parental behavior may have a greater influence on children's motivational beliefs. Because this study focused on the differential effects of maternal and paternal beliefs and behaviors, it seemed reasonable to consider both parents separately. However, subsequent studies could examine the interaction of both parents, for example, by examining the effects of opposing or coordinated parental beliefs and behaviors [64].

## Conclusion

Studies suggest that girls hold a lower math self-concept of ability (MSC) compared to boys [3–6]. These gender differences were thought to contribute to disparities in students' educational and career choices [7–11]. This suggests a need to examine the mechanisms that contribute to children's MSC. Based on the Parental Socialization Model, our study expands the prior literature by testing the links between mothers' and fathers' own MSC, their child-specific performance expectations, and their encouragement regarding math and science activities in elementary school on boys' and girls' MSC in both elementary and secondary school. While we found no gender differences in parental expectations and encouragement of girls and boys, we found different mechanisms that contribute to their MSCs.

Overall, the results of this study suggest that mothers and fathers tended to translate their own MSC into their mathematical performance expectations and encouragement of their sons, but not their daughters. This mechanism could contribute to the maintenance of gender differences, as parental expectations, especially father's performance expectations, have been shown to have an important effect on children's MSC. These findings suggest that frameworks for parental socialization do not apply in the same way for boys and girls and that a differentiation according to child's gender is highly relevant and should therefore be considered in follow-up studies.

Furthermore, we investigated the role of parents own MSC. Our results suggest that only fathers' MSC was indirectly related to sons' MSC via performance expectations. This suggests that fathers who view themselves as capable in mathematics are more likely to expect similar abilities in their sons, thereby fostering their sons' confidence in math. Contrary to our hypothesis, however, we found that the effects of fathers' MSC, compared to the effects of mothers MSC, on sons' MSC were not significantly different. This means that although we find an indirect effect of the fathers' MSC on the sons' MSC, this effect is not significantly greater than in the mother-son dyad. Considering the small sample sizes in the dyads, this could be due to the low power of the study, which is why this hypothesis should be tested in follow-up studies with larger samples. Furthermore, both boys' and girls' MSC were predicted by their fathers' performance expectations and girls' MSC by their fathers' math-specific encouragement.

Our findings highlight the relevance of fathers' early educational encouragement of math and science activities and may lead to effective interventions to improve boys' and girls' mathematical competence beliefs. The mere knowledge of what role fathers play in the development of their children' MSC, could result in a change in their beliefs and behaviors. As studies show, that fathers' school- and home-based involvement is lower than mothers' (for an overview see [65]), interventions should raise awareness among fathers and seek to increase their educational participation. Future studies should build on our findings and investigate gendered parent-child mechanisms in other subjects and domains. Furthermore, to place our findings in a larger picture, different parental behaviors should be taken into account.

## Supporting information

**S1 Table. Testing measurement invariance across time and groups in measures of children's math self-concept of ability.** $N = 517$, all $\chi^2$ values are statistically significant with $p <$ .01. To evaluate $\Delta$CFI, Models 2 and 3 were compared with Model 1. TLI = Tucker-Lewis index, CFI = comparative fit index, RMSEA = root mean square error of approximation, 90% CI = 90% confidence interval RMSEA, SRMR = standardized root mean squared residual. (DOCX)

**S2 Table. Mean comparisons included and excluded study participants.** MSC = math self-concept of ability; included, $n = 517$; excluded, $n = 482$. *$p < .05$. ***$p < .001$ (two-tailed). (DOCX)

**S3 Table. Unstandardized and standardized path coefficients for stability coefficients and covariates (continuation of Table 3).** UNSTD = unstandardized model results, MSC = math self-concept of ability, F/M = mother/father, MSC = math self-concept of ability, M/F = mother/father, grade = math grade w1. [a]Arrow indicates direction of causation. *$p < .05$. **$p < .01$. ***$p < .001$. (DOCX)

## Author Contributions

**Conceptualization:** Paulina Feige, Rainer Watermann, Elisa Oppermann.

**Data curation:** Sandra Simpkins, Jacquelynne Sue Eccles.

**Formal analysis:** Paulina Feige.

**Funding acquisition:** Jacquelynne Sue Eccles.

**Methodology:** Rainer Watermann, Elisa Oppermann.

**Supervision:** Rainer Watermann, Sandra Simpkins, Jacquelynne Sue Eccles, Elisa Oppermann.

**Validation:** Rainer Watermann, Elisa Oppermann.

**Writing – original draft:** Paulina Feige.

**Writing – review & editing:** Rainer Watermann, Elisa Oppermann.

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
