## [Decision Letter · Decision Letter 0]

28 Oct 2024

PONE-D-24-20457Impact of Mothers’ and Fathers’ Math Self-Concept of Ability, Child-Specific Beliefs and Behaviors on Girls’ and Boys’ Math Self-concept of AbilityPLOS ONE

Dear Dr. Feige,

Thank you for submitting your manuscript to PLOS ONE. After careful consideration, we feel that it has merit but does not fully meet PLOS ONE’s publication criteria as it currently stands. Therefore, we invite you to submit a revised version of the manuscript that addresses the points raised during the review process.

After a thorough review of the evaluations, it is clear that while the topic addressed is highly relevant and innovative, the manuscript requires significant revisions. In particular, the theoretical framework needs to be expanded to better position the study within the existing literature, and the methodological choices need to be more rigorously justified. These revisions will allow the study to better realize its potential.

We look forward to receiving your revised manuscript.

Kind regards,

Leonard Moulin

Academic Editor

PLOS ONE

2. Thank you for stating the following financial disclosure: [The Childhood and Beyond Study was funded by the National

Institute for Child Health and Human Development, Grant HD17553]. Please state what role the funders took in the study. If the funders had no role, please state: "The funders had no role in study design, data collection and analysis, decision to publish, or preparation of the manuscript." If this statement is not correct you must amend it as needed. Please include this amended Role of Funder statement in your cover letter; we will change the online submission form on your behalf.

Additional Editor Comments (if provided):

Reviewers' comments:

Reviewer's Responses to Questions

**Comments to the Author**

1. Is the manuscript technically sound, and do the data support the conclusions?

Reviewer #1: Yes

Reviewer #2: Partly

2. Has the statistical analysis been performed appropriately and rigorously? 

Reviewer #1: Yes

Reviewer #2: Yes

3. Have the authors made all data underlying the findings in their manuscript fully available?

Reviewer #1: Yes

Reviewer #2: Yes

4. Is the manuscript presented in an intelligible fashion and written in standard English?

Reviewer #1: Yes

Reviewer #2: Yes

5. Review Comments to the Author

Reviewer #1: The current study analyses the question of the extent to which maternal and paternal

math ability self-concept, their child-specific math performance expectations and

encouragement of math at home differentially impact the mathematical self-concept of their children. Therefore, a mediation model was tested in a sample of N = 517 children and their parents from a longitudinal study from the years 1986 to 1994. An important result was that only fathers’ performance expectations were relevant for boys’ and girls’ self-concept in mathematics. Moreover, fathers’ encouragement was relevant for girls’ self-concept in math.

Thank you for the opportunity to review this manuscript. In my opinion, the topic is highly relevant. After all, parents have a major influence on the education of their children. As there are hardly any studies on the influence of parental school self-concept on children's assessments, the present study is a very good addition to the research situation. The theory section is very well structured and clearly written. The relevant studies are listed and the research gaps are pointed out. The statistical analyses are clearly presented and appropriate to the questions posed.

However, I would just like to give the authors some ideas for thought as to what other points could be addressed. For example, I find it difficult to focus solely on the influence of parents. The school and the teachers could play an equally important role when it comes to the development of the academic self-concept. Is there any possibility of integrating the school context into the present analyses? In principle, of course, studies on the heritability of academic self-concept would be important - are there already any studies that have investigated the genetic influence and the influence of the environment? If not, this aspect could also be taken up in the discussion. As the authors stated themselves, it could also be that the results are specific to the recruitment period (up to the 1990s). It is possible that a lot has changed in the last 30 years in terms of parents' self-perceptions of ability and expectations. Maybe the authors could provide more information to validate their results.

Here are some minor issues that could be addressed additionally:

• It would have been desirable to provide more than one item to record the parents‘ self-concepts.

• As the missing values were relatively high for the fathers, it might be helpful to take a look at the analyses without the missings and compare the results of these analyses. Although FIML was used to account for the missing values – I think it could be important to provide additional analyses here in an attachment.

• It seems to me that only families with both parents were investigated. What about single parents? Were these parents eliminated from the sample?

• I think it would be helpful to get more information about the sample. For example, for what purpose the families took part in the study and how they were recruited.

• I have been thinking about whether it would make sense to shed light on the discrepancy in expectations between mothers and fathers. Do you think it could be that the parents view their child differently and this discrepancy is relevant for the MSC of the children?

Reviewer #2: The manuscript "Impact of Mothers’ and Fathers’ Math Self-Concept of Ability, Child-Specific Beliefs and

Behaviors on Girls’ and Boys’ Math Self-concept of Ability” investigates the indirect

relations between mothers’ and fathers’ MSC and boys’ and girls’ MSC, mediated by

parents’ child-specific math performance expectations, and parents’ encouragement regarding

math and science activities in 517 children and their parents.

The topic is relevant although, in my opinion, the contribution currently fails to fully exploit this potential; I would therefore suggest reconsidering the article for publication should the authors manage to make the appropriate changes.

#1 - Introduction

This section discusses several key concepts that would require appropriate elaboration.

- "“several theories emphasize the importance of parental beliefs and

behaviors for the development of school-relevant outcomes in children”: this aspect should be expanded and reference should also be made to other significant adults whose beliefs may have an impact on learning e.g. teachers.

With respect to the theoretical framework, the reader is not adequately guided to the theoretical interpretation of the processes involved in learning. This requires a considerable expansion of the specific literature (Parental Socialisation Model).

- “extensive research has shown that there is a positive relation

between mothers’ and fathers’ child-specific math performance expectations and children’s

MSC for different samples with respect to age and nationality”: this expression is not enough, some more relevant work on this topic is needed.

- “there is evidence that parents’ math anxiety is transferred to their

children, with stronger effects in same-sex dyads”: this sentence is very disconnected from the rest and yet it is not even deepened; it is necessary to avoid these situations by expanding and rewriting the entire period.

- "children seem to adopt their parents’ interpretations of reality and integrate

them into their own self-related cognitions": this part is very interesting it needs to be expanded and justified with scientific studies.

Finally, the introduction does not make explicit the ‘gap’ in the literature that the study will fill. Is it the evolution over the life span? The implications in reference to performance? It might prove useful to better integrate this aspect with the results already available in the literature.

#2 Method

I have major doubts about this section; the first aspect concerns recruitment and the significant time elapsed since the initial assessment, which for obvious reasons photographs a condition that may not fully represent current functioning both from the point of view of the socio-affective processes involved and the cognitive performance measured.

I also believe that the absence of ethical approval may represent a relevant constraint for the journal.

Another relevant issue relates to the detection of constructs; I find it methodologically flawed to detect parental beliefs by means of a rating item only. This may not seem to capture the conceptual complexity of what is being measured. It is unclear to me why the author(s) did not make use of objective assessment instruments that would certainly have captured the research interest more effectively. This should be justified in the manuscript.

The statistical analyses carried out seem adequate but nevertheless suffer greatly from these arbitrary measurements and therefore risk losing their meaning. The use of this type of measurement should certainly be adequately justified.

The conclusions are not appropriate for the scientific product; this section should be completely revised by expanding the relations between the constructs in light of the results obtained. The entire dimension related to the results of the study and the findings should also be stressed.

6. PLOS authors have the option to publish the peer review history of their article (what does this mean?). If published, this will include your full peer review and any attached files.

Reviewer #1: No

Reviewer #2: No

---

## [Author Response · Author response to Decision Letter 0]

4 Dec 2024

Comment Answer Excerpt from revised manuscript

Please ensure that your manuscript meets PLOS ONE's style requirements, including those for file naming. Thank you for the suggestion, the manuscript has been revised accordingly. 

Please state what role the funders took in the study. If the funders had no role, please state: "The funders had no role in study design, data collection and analysis, decision to publish, or preparation of the manuscript." The funders had no role in the study and we added this sentence to the resubmission letter. 

Please include your full ethics statement in the ‘Methods’ section of your manuscript file. In your statement, please include the full name of the IRB or ethics committee who approved or waived your study, as well as whether or not you obtained informed written or verbal consent. If consent was waived for your study, please include this information in your statement as well. We included the following statements in the ‘Method’ section. 

 Method:

Parents provided written informed consent prior to their children's participation. A letter outlining the study details was sent home with the children, which included an informed consent form for the parents to sign. The signed consent forms were then returned via the children to their teachers. As an incentive for participation, the researchers informed the schools that the population-level results would be shared with them, and that each participating teacher/classroom would receive a small financial contribution, which could be used to purchase educational materials. Data collection was carried out in the classrooms only for those students whose parents had returned the signed consent form. (pp.10-11)

[…]

The Ethics Committee of Freie Universität Berlin has confirmed that no ethical approval is required. (p.11)

Please include a separate caption for each figure in your manuscript. Thank you for the suggestion, we included a separate caption for each figure in the manuscript. 

Reviewer 1: Is there any possibility of integrating the school context into the present analyses?

Reviewer 2: This section discusses several key concepts that would require appropriate elaboration.

- "“several theories emphasize the importance of parental beliefs and

behaviors for the development of school-relevant outcomes in children”: this aspect should be expanded and reference should also be made to other significant adults whose beliefs may have an impact on learning e.g. teachers. Thanks for the valuable suggestions. We have taken this into account at various points in the ‘Introduction’ and ‘Limitations’ section. Introduction: 

Because parents are considered to be the most important agents of primary socialization [16], several theories emphasize the importance of parental beliefs and behaviors for the development of school-relevant outcomes in children (e.g., [17, 18]). For example, within the Bioecological Model of Human Development [17], the immediate social environment, the so-called microsystem, provides a crucial context for human development. Especially in the early years, parents are considered the principal persons within the microsystem; however, over time, interactions with significant others—such as teachers and peers—also influence a child’s development. While acknowledging especially the impact of teacher’s beliefs and behaviors on a child’s MSC [19, 20] this paper focuses on parents as the primary socializing agents. (pp. 3-4)

Limitations: 

Eighth, as initially mentioned, there are a variety of processes that are involved in the development of ability self-concepts [12, 13]. In order to explore the relative importance of different processes (e.g. interactions with teachers, comparisons with peers and across domains) in the formation of the academic self-concept, follow-up studies should focus on different mechanisms simultaneously. (pp. 31-32)

Reviewer 1: In principle, of course, studies on the heritability of academic self-concept would be important - are there already any studies that have investigated the genetic influence and the influence of the environment? If not, this aspect could also be taken up in the discussion. Thank you for the valuable suggestion. We discussed this point in the ‘Limitations’ section. Limitations: 

Seventh, when studying effects within families, genetic factors must also be considered. Since our study assumed environmental factors within families as mediating processes between parents and child’s MSC, one can also assume that there might be a genetic influence. Twin studies have shown that individual differences in self-perceived abilities can be explained by both genetic factors and non-shared environments [62, 63]. Although the data in this study did not allow us to account for a genetic component, it is important to consider the heritability of academic self-concept when interpreting our findings. (p. 31)

Reviewer 1: As the authors stated themselves, it could also be that the results are specific to the recruitment period (up to the 1990s). It is possible that a lot has changed in the last 30 years in terms of parents' self-perceptions of ability and expectations. Maybe the authors could provide more information to validate their results.

Reviewer 2: I have major doubts about this section; the first aspect concerns recruitment and the significant time elapsed since the initial assessment, which for obvious reasons photographs a condition that may not fully represent current functioning both from the point of view of the socio-affective processes involved and the cognitive performance measured. Thanks for the helpful comments. We have elaborated on this point in the limitations-section and now report stronger empirical evidence to validate our results. Limitations: 

Fourth, it should be noted that the CAB study used data from 1988-1995, which means that the results of this study may not be fully transferable to children the present day. However, the participants in our study are now in their forties, so knowledge of their motivational beliefs during their childhood is still relevant. Furthermore, current studies still reveal gender differences in mathematical ability perceptions of boys and girls [12] and mothers and fathers [60] and parental gender stereotypic beliefs and expectations in math [3, 4, 45]. For example, Starr et al. [45] compared datasets from 1984 to 2009 and found that parents had gender stereotyped beliefs in math across data sets. Furthermore, the importance of parental performance expectations for children’s MSC were also found in a current study [26]. Thus, it is still highly relevant to gain a better understanding of the early mechanism that eventually contribute to these gender difference in math-related beliefs. (pp. 31-32)

Reviewer 1: It would have been desirable to provide more than one item to record the parents‘ self-concepts.

Reviewer 2: Another relevant issue relates to the detection of constructs; I find it methodologically flawed to detect parental beliefs by means of a rating item only. This may not seem to capture the conceptual complexity of what is being measured. It is unclear to me why the author(s) did not make use of objective assessment instruments that would certainly have captured the research interest more effectively. This should be justified in the manuscript.

The statistical analyses carried out seem adequate but nevertheless suffer greatly from these arbitrary measurements and therefore risk losing their meaning. The use of this type of measurement should certainly be adequately justified. We have also responded to this point in the ‘Method’ and ‘Limitations’ section. Method: 

Parents’ child-specific mathematical performance expectations were operationalized using the item “How well do you think this child will do in each of these areas in the next year? Please use this scale: -Math”. Responses could be given on a 7-point Likert response format from Not at all well (1) to Very well (7). With regard to the criterion validity of the item, correlation patterns with grades are comparable to those found in Pesu et al [26] with multi-item scales (Pesu et al. [26]: rmother = 41, rfather = 31; in our study: rmother = 36; rfather = 31, Table S3). (pp. 12-13)

Limitations: 

Fifth, parental MSC, their performance expectations and their encouragement were measured with single items. Therefore the reliability of the constructs could not be tested. As stated in the methods section, results of Gogol et al. [47] suggested that single items measures of academic self-concepts meet both validity and reliability criteria. Furthermore, there is evidence that most single-item measures are as valid and reliable as multi-item scales [60]. With regard to performance expectations, we provided additional evidence for criterion validity in the methods section. Moreover, the significant associations of these single items with children’s MSC suggest that these measure meaningful aspects of parental beliefs and practices. (p. 31) 

Reviewer 1: As the missing values were relatively high for the fathers, it might be helpful to take a look at the analyses without the missings and compare the results of these analyses. Although FIML was used to account for the missing values – I think it could be important to provide additional analyses here in an attachment. Thank you for this suggestion. We have provided additional information explaining why we believe that FIML is the best approach for handling high percentages of missing data. We also discussed the possibility of providing additional analyses using listwise deletion. We are willing to do this if desired; however, it raises the question of what conclusions should be drawn if, for example, the results differ: “This prohibition should also extend to the intuitive yet misguided practice of using listwise deletion to ‘‘double check’’ the

accuracy of more robust missing data approaches like ML estimation and multiple imputation. There is no good reason for this. When listwise deletion yields discrepant results from ML or MI techniques, this does not in any way cast doubt onto the ML and MI results; rather, it only suggests that

the missingness mechanism is in part MAR (which is quite often the case). Further, there is no logical basis for using listwise deletion in this way. If the listwise result agrees with the ML and MI result, then we will accept the ML and MI result; and if the listwise result disagrees with the ML and MI result, then we will still accept the ML and MI result (because ML and MI provide accurate SEs and are unbiased under both MAR and MCAR mechanisms, whereas listwise deletion provides highly inflated SEs and is only unbiased under MCAR)—the information value of the listwise dele-tion result is nil either way. ” (Newman, 2014, pp. 384-385). 

Reference:

Newman DA. Missing data: Five practical guidelines. Organizational Research Methods. 2014;17(4):372–411. doi:10.1177/1094428114548590 Method: 

Model parameters were estimated using the FIML algorithm [52]. Studies suggest that FIML outperforms many techniques for handling missing data (e.g. listwise deletion) when the percentage of missing values is relatively high (e.g. 50%; [52, 53]). (p. 15)

Reviewer 1: It seems to me that only families with both parents were investigated. What about single parents? Were these parents eliminated from the sample? As stated in the ‘Method’ section, only children who had at least one parent (mother or father) participating in w1 were included in our study. To avoid any misunderstandings, we provided additional information about the current marital status as reported by mothers.

 Method: 

Only children who participated in w1 in the study and who had at least one parent (mother or father) participating in w1 were included in our study (n = 517). (p. 11)

[…]

In w1, 0.6% of mothers stated that they were single, 89.7% were married, 1.7% were separated, 5% were divorced, and 0.6 % were widowed. (p. 12)

Reviewer 1: I think it would be helpful to get more information about the sample. For example, for what purpose the families took part in the study and how they were recruited. Thank you for this comment, we added information of the sample in the ‘Method’ section. Method: 

We draw on data from 517 children and their mothers and fathers from all three cohorts of the CAB Study. CAB is a large-scale longitudinal study that included 10 data collection points, which began in the school year 1986/1987 in four school districts in Southeastern Michigan with three different age cohorts. Recruitment of participants started on December 1st, 1986 and ended October 30th, 1988. Parents provided written informed consent prior to their children's participation. A letter outlining the study details was sent home with the children, which included an informed consent form for the parents to sign. The signed consent forms were then returned via the children to their teachers. As an incentive for participation, the researchers informed the schools that the population-level results would be shared with them, and that each participating teacher/classroom would receive a small financial contribution, which could be used to purchase educational materials. Data collection was carried out in the classrooms only for those students whose parents had returned the signed consent form. (pp. 10-11)

Reviewer 1: I have been thinking about whether it would make sense to shed light on the discrepancy in expectations between mothers and fathers. Do you think it could be that the parents view their child differently and this discrepancy is relevant for the MSC of the children? Thank you for this interesting suggestion. To address the question, whether parents may view their child differently, we added paired t-tests to compare mothers’ vs. fathers’ expectations and their encouragement. 

How this discrepancy affects childrens’ MSC is also an interesting question. However, we do not think we can answer this point with our research design, as we focus on parent-child dyads (e.g. mothers and daughters) in our study. To address this question, we would have to look at both parents within the same model and possibly form groups according to agreement (e.g. both parents high, only one parent high, both parents low; as in Simpkins et al., 2005). However, we have included this point in the discussion. 

Reference: 

Simpkins SD, Davis-Kean PE, Eccles JS. Parents' socializing behavior and children's participation in math, science, and computer out-of-school activities. Applied Developmental Science. 2005;9(1):14–30. doi:10.1207/s1532480xads0901_3 Results: 

Additionally, compared to fathers, mothers reported having higher expectations for their daughters’ math performance, t(155) = 2.51, p = .013, with a small effect size, d = .20, 95% CI for d [.04, .36]. In contrast, no difference was found between mothers and fathers in how much they encouraged their daughters, t(156) = 1.24, p = .218. For sons, there were no significant differences between mothers and fathers regarding either their performance expectations, t(162) = 0.69, p = .493, or their encouragement, t(158) = 0.81, p = .421. (p. 18)

Limitations: 

Ninth, it can be assumed that maternal and paternal socialization do not operate independently, as our model implies. Thus, coordinated parental behavior may have a greater influence on children’s motivational beliefs. Because this study focused on the differential effects of maternal and paternal beliefs and behaviors, it seemed reasonable to consider both parents separately. However, subsequent studies could examine the interaction of both parents, for example, by examining the effects of opposing or coordinated parental beliefs and behaviors [65]. (p. 32)

Reviewer 2: With respect to the theoretical framework, the reader is not adequately guided to the theoretical interpretation of the processes involved in learning. This requires a considerable expansion of the specific literature (Parental Socialisation Model).

"children seem to adopt their parents’ interpretations of reality and integrate

them into their own self-related cognitions": this part is very interesting it needs to be expanded and justified with scientific studies. We added some more information about the model and the lear

---

## [Editor Report · Decision Letter 1]

7 Jan 2025

Impact of Mothers’ and Fathers’ Math Self-Concept of Ability, Child-Specific Beliefs and Behaviors on Girls’ and Boys’ Math Self-concept of Ability

PONE-D-24-20457R1

Dear Dr. Feige,

We’re pleased to inform you that your manuscript has been judged scientifically suitable for publication and will be formally accepted for publication once it meets all outstanding technical requirements.

Kind regards,

Leonard Moulin

Academic Editor

PLOS ONE
---

## [Editor Report · Acceptance letter]

13 Jan 2025

PONE-D-24-20457R1 

PLOS ONE

Dear Dr. Feige, 

I'm pleased to inform you that your manuscript has been deemed suitable for publication in PLOS ONE. Congratulations! Your manuscript is now being handed over to our production team.

Kind regards, 

on behalf of

Dr. Leonard Moulin 

Academic Editor

PLOS ONE